# Physical activity among older adults with multimorbidity: Evidence from a population-based health survey

**Bruno Holanda Ferreira**[1☉]*, **Ricardo Goes de Aguiar**[2☉], **Edige Felipe de Sousa Santos**[2☉], **Chester Luiz Galvão Cesar**[2☉], **Moisés Goldbaum**[1☉], **Camila Nascimento Monteiro**[1,3☉]

1 Departamento de Medicina Preventiva, Faculdade de Medicina, Universidade de São Paulo, São Paulo (SP), Brazil, 2 Departamento de Epidemiologia, Faculdade de Saúde Pública, Universidade de São Paulo, São Paulo (SP), Brazil, 3 Hospital Sírio-Libanês, São Paulo (SP), Brazil

☉ These authors contributed equally to this work.
* holanda19@hotmail.com

## Abstract

### Introduction

The promotion of physical activity has been recognized as an important component in the management and prevention of multimorbidity, a condition that is increasing prevalent worldwide, including in Brazil. However, there is a scarcity of studies exploring the disparity in physical activity levels between individuals with and without multimorbidity. Therefore, the study aimed to estimate the prevalence of multimorbidity and physical activity among older adults, as well as analyze the relationship of a sufficient level of physical activity and multimorbidity, while considering sociodemographic characteristics of residents in São Paulo, Brazil.

### Materials and methods

Data from 1.019 participants aged 60 years or older (59.7% female; mean age 69.7±7.7) were collected from the Health Survey (ISA-Capital, 2015) conducted in the city of São Paulo, Brazil. We defined multimorbidity as the presence of two or more chronic conditions, and for physical activity, classified a sufficient level ($\geq$150 min/week). Prevalence Ratios (PR) with 95% Confidence Intervals (95%CI) were estimated using univariate and multivariate Poisson regression to examine the relationship between multimorbidity and sufficient level of physical activity.

### Results

67.7% of the participants lived with multimorbidity, while 30.1% had achieved a sufficient level of physical activity. There was a higher prevalence of sufficient level of physical activity among older adults with two (PR = 1.38; 95%CI 1.02–1.88) and four (PR = 1.37; 95%CI 1.00–1.87) chronic conditions. Older adults with multimorbidity who were 70 years or older (PR = 1.77; 95%IC 1.13–2.77), female (PR = 1.65; 95%CI 1.16–2.36), without a partner (PR = 1.43; 95%IC 1.03–1.99), and had a per capita income of 1 to 2.5 (PR = 1.83; 95%IC 1.00–

**Data Availability Statement:** All relevant data are within the manuscript and its Supporting Information files.

**Funding:** BHF; Coordenação de Aperfeiçoamento de Pessoal de Nível Superior – Brasil (CAPES); Número do processo: 88887.505957/2020-00; Código Financeiro 001; https://www.gov.br/capes/pt-br FEFS; Supported by the São Paulo Research Foundation (FAPESP), which financed the postdoctoral scholarship (grant n° 2020/03013-0); https://fapesp.br/.

**Competing interests:** The authors have declared that no competing interests exist.

3.33) were more likely to achieve a sufficient level of physical activity compared to their peers without multimorbidity.

## Conclusions

The study highlights sociodemographic disparities in the sufficient level of physical activity among multimorbidity, suggesting the importance of considering these factors when planning public policies aimed at promoting physical activity.

## Introduction

The global population of older adults has been rapidly increasing because of advancements in the treatment and management of various diseases. It is projected that, by 2050, approximately 22% of the world's population will be aged 60 years or older [1], with an estimated 14% in Brazil by 2035 [2]. Despite this significant demographic shift, a substantial number of older adults are affected by multimorbidity, which refers to the simultaneous occurrence of two or more chronic health conditions [3, 4]. The prevalence of multimorbidity among older adults has been reported as 31.4% in Europe [5], 33.3% in Latin America and the Caribbean [3], and 42.9% in Brazil [6]. Multimorbidity is considered a public health issue [7] and is associated with the current demographic and epidemiological transition [8], aging [9–11], increased mortality rates [12], reduced life expectancy [13], sociodemographic inequalities [11], reduced functional capacity and physical inactivity [14, 15], negative self-rated health [14, 16, 17] decreased quality of life [18], and a higher likelihood of seeking health care [19].

Emerging paradigms in health promotion and disease prevention emphasize various strategies, with a particular focus on fostering and sustaining an active lifestyle that involves accumulating physical activity over the course of a week. Research has demonstrated that being physically active (defined as practicing at least 150 minutes of physical activity per week) is associated with a higher likelihood of healthy aging, and a reduced risk of non-communicable diseases (NCD's) [20–22], including depression, hypertension, diabetes [23], multimorbidity and high medication consumption [24]. Furthermore, being physically active is associated to a decreased risk of mortality [20] and disability [25]. However, it is observed that a significant proportion of adults and older adults do not engage in sufficient physical activity worldwide, with approximately 21.4% falling into this category [26] and the prevalence is even higher at 72.5% among older adults in Brazil [27]. It is estimated that physical inactivity leads to approximately 3.2 million deaths annually [28]. Therefore, it is plausible to recommend promoting increased physical activity to mitigate the complications that arise from multimorbidity, therefore reducing mortality among older adults.

Studies have highlighted that the leisure and commuting domains of physical activity have an inverse relationship with all causes of disease. In a cohort study, Feter et al. [29] examined the relationship between leisure-time physical activity and multimorbidity. The researchers identified a protective effect of physical activity against multimorbidity. A longitudinal investigation by Zhou et al. [30], showed that older adults who performed both regular and irregular physical activity were significantly associated with the late onset of chronic diseases such as hypertension, hyperlipidemia, diabetes, cardiovascular diseases, renal failure, liver and biliary system diseases, overweight and obesity.

However, to assist in proposing actions and public policies for health promotion and prevention that particularly address behavior among individuals who practice physical activity to

reduce the risk of chronic diseases, as recommended by the American Society of Geriatrics [31], physical activity guide for the Brazilian population [32] and Guidelines for Physical Activity and Sedentary Behavior [33]. Given the increasing concern about physical activity among older adults with multimorbidity, several studies have examined the prevalence of physical activity among older adults with multimorbidity [14, 27, 34]. However, it is essential to analyze the behavior of older adults with or without multimorbidity who reach a sufficient load of physical activity characteristic that is still little explored low-and-middle-income countries like Brazil.

Examining the behavior towards physical activity among older adults, both with and without multiple chronic health conditions, and exploring the sociodemographic distinctions between these groups, fill a gap in the literature. Additionally, we are convinced that this comprehension has significant implications for shaping public policies and subsidizing the implementation of interventions geared towards promoting physical activity. The present study aims to estimate the prevalence of multimorbidity and physical activity among older adults living in a Latin America megacity, and to analyze the relationship of a sufficient level of physical activity and multimorbidity, according to their socioeconomic and demographic characteristics.

## Materials and methods

This is a cross-sectional, population-based study that used data from the Health Survey in the City of São Paulo of 2015, in which the sample studied refers to the population living in an urban area of the city of São Paulo, southeastern Brazil. The city of São Paulo has more than 11 million inhabitants, and about 12% are older adults. It is the largest city in Brazil, the most economically developed, and the fourth-largest metropolis in the world in terms of population [35, 36].

The 2015 ISA-Capital was held to evaluate the health status of São Paulo City population according to their living conditions, and to address indicators of lifestyle and chronic diseases, detailed information in another study [37]. The sampling was collected by following a complex and probabilistic methodology, drawing data from census sectors and households. Two domains, geographical and demographic, were considered, covering, respectively, regional health coordinating body (Midwest, East, North, Southeast, and South) and age range (males and females with 60 years or older). The total sample of 980 individuals was established, and, for each regional body, 162 to 234 interviews would be conducted. This allowed us to estimate proportions of 0.50, with a sampling error of 0.10, considering a confidence level of 95% and a design effect of 1.5 [37].

The data was collected for the present study was conducted between January and December 2015, before the COVID-19 pandemic. During this period, selected households were visited at least three times to ensure data collection. One important characteristic if the ISA-Capital study is non-use of intra-household selection. Considering a total sample, the response rate per household was 0.76, while the response rate of interviews with the eligible population in these households was 0.74, indicating a high level of cooperation from the individuals being interviewed. A detailed description of the ISA-Capital 2015 sampling plan of was the subject of another publication [37].

Participants were recruited considering that they were within the contemplated age range and lived in permanent regime in private urban households. Institutionalized people and homeless individuals were excluded. Information was obtained through questionnaires applied by trained interviewers and answered directly by 60-year and older male and female residents. The questionnaire was organized into thematic blocks, whose questions were mostly

close-ended or presented predefined alternatives. All participants included in the current analysis provided data on health condition, demographic information, socioeconomic status, and physical activity. It is important to highlight that the researchers involved in this study did not have access to any personal data that could potentially identify individual participants during or after the data collection process.

Following multimorbidity definition standards [3,4], participants were divided into two groups: a) Without Multimorbidity (had zero or one health condition); b) With Multimorbidity (had two or more chronic health conditions [3]), based on self-reported diagnosis of conditions from the following list: hypertension, diabetes, angina, myocardial infarction, cardiac arrhythmia, other heart diseases, cancer, arthritis, rheumatism, arthrosis, osteoporosis, asthma or asthmatic bronchitis, emphysema, chronic bronchitis or chronic obstructive pulmonary disease, rhinitis, chronic sinusitis, other lung diseases, tendinitis, repetitive strain injury, work-related musculoskeletal disorder, lower limbs varicose veins, cerebrovascular accident or stroke, other diseases that affect the veins, arteries or blood circulation, cholesterol, spinal cord disorders or spinal problems, mood disorders or mental illnesses such as anxiety, depression, panic disorder, obsessive-compulsive disorder, schizophrenia, or any other chronic diseases other than those previously mentioned. Except for mental illnesses, in which subjects were asked: "Do you have any kind of mood disorder or mental illness such as anxiety depression, panic disorder, obsessive-compulsive disorder, schizophrenia, or other?" For other conditions, participants were asked: "Has any doctor ever told you that you have [name of the disease]?". In this variable, there was no missing information.

Considering the relevance to public health [4], we included the following independent variables: age range in complete years (60–69 or ≥70); gender (male or female); race/skin color (white or non-white); marital status (with a partner or without a partner); education (never been to school; up to elementary school; up to junior high-school; up to high-school; up to college; up to graduate studies); occupation or volunteer work (yes; no); income per capita (≤0.5; 0.5–1.0; 1–2.5; ≥2.5) and number of chronic conditions (0–1; 2; 3; 4; ≥5). The variables race/skin color, marital status, and occupation or volunteer work contained incomplete information and were treated as missing data.

Physical activity was considered a dependent variable according to leisure-time and commuting physical activity. The information was retrieved by using the long version of the International Physical Activity Questionnaire (IPAQ), which has been validated [38, 39] and explored in other studies relying on ISA-Capital data [40–43]. The questions referred to weekly frequency and total daily time spent in physical activity. For example: "In a typical week, on how many days do you engage in vigorous physical activity for at least 10 minutes?" Responses ranged from 1 to 7 days. Also, "How much time do you usually spend in vigorous physical activities on one of these days?". The scores were transformed into minutes/week and were calculated according to the international guidelines of the IPAQ (www.ipaq.ki.se) considering the multiplying of the frequency of vigorous physical activities by two [40]. The time spent on leisure and commuting physical activity was categorized as either 0–150 minutes/week, which was considered an insufficient level of physical activity or ≥150 minutes/week as sufficient level of physical activity. This categorization was chosen based on the guidelines for the practice of physical activity [32]. After processing the data, incomplete information was identified and subsequently treated as missing data (See data in S1 File).

The descriptive analysis of sociodemographic characteristics, chronic health conditions, multimorbidity, and physical activity level of the total sample was estimated using prevalence and a 95% Confidence Interval (95%CI). Chi-Square was employed to analyze the relationship between physical activity and the number of chronic health conditions. Considering that this is a cross-sectional study, with a binary outcome variable and high prevalence (>10%), Poisson

Regression is preferable to logistic regression as it provides prevalence estimates with more conservative confidence intervals [44]. Therefore, we estimated the Prevalence Ratio (PR) and 95%CI with robust variance to identify differences in the sufficient level of physical activity with the number of chronic health conditions, and also to identify differences between the sufficient level of physical activity among older adults with and without multimorbidity according to their sociodemographic characteristics using a univariate model. Poisson's multiple regression model estimates were also used by adjusting for age, education, and gender. A significance level of 5% was adopted for all analyses.

The Stata statistical package (Stata, version 14, *StataCorp*, *College Station*, Texas, USA) was used, which allows the inclusion of aspects related to the complex sample design, namely strata, conglomerates, and weightings, through the *survey* module.

All procedures performed in studies involving human participants were in accordance with the ethical standards of the institutional and/or national research committee. The study is consistent with Resolution No. 466/12 of the Brazilian National Health Council. Informed consent was obtained from all participants after a full explanation of the purpose and procedures of the research. All participants signed the Free and Informed Consent Form. This study was approved by the Ethics Committee of the Faculdade de Saúde Pública, Universidade de São Paulo–São Paulo (SP), Brazil (protocol: 719,661/2014).

## Results

Table 1 shows the characteristics of the study sample (n = 1.019). Considering the due sample weights, most of the older adults in the sample were within the 60–69 age range, females, white race/skin color, low educational level, with no occupation or volunteer work, low per capita income, 0–1 chronic health conditions, with multimorbidity and insufficient level of physical activity.

The results of the regression analysis between the frequency of chronic health conditions and physical activity show that older adults who have two chronic conditions (PR = 1.38; 95% CI 1.02–1.88; p = 0.039) and four chronic conditions (PR = 1.37; 95%CI 1.00–1.87; p = 0.049) have increased likelihood of having a sufficient level of physical activity, while the state of physical activity was maintained for individuals with three and five or more chronic conditions, when compared to older adults with up to one chronic condition; Table 2.

Table 3 shows the distribution sufficient level of physical activity among older adults with and without multimorbidity according to socioeconomic, demographic characteristics and regression analysis. The crude analysis evidenced an association between physical activity and multimorbidity among older adults aged 70 years or more and female (p<0.050). After adjusting the model, it was observed that the age group of 70 years or more and female continued to exhibit significance, and significance was also found for older adults who do not live with a partner and those with a per capita income of 1 to 2.5, (p<0.050). Thus, individuals with multimorbidity who are in the 70-year-old group or older (PR = 1.77; 95%IC 1.13–2.77; p = 0.013), female (PR = 1.65; 95%IC 1.16–2.36; p = 0.006), do not have a partner (PR = 1.43; 95%IC 1.03–1.99; p = 0.032), and have a per capita income of 1 to 2.5 (PR = 1.83; 95%IC 1.00–3.33; p = 0.047) present higher likelihood of achieving a sufficient level of physical activity when compared to their peers without multimorbidity and with a sufficient level of physical activity.

## Discussion

A high prevalence of multimorbidity and a low prevalence of a sufficient level of physical activity were identified among older adults residents in the city of São Paulo, Brazil. A higher

**Table 1. Distribution of demographic and socioeconomic characteristics, number of chronic health conditions, multimorbidity, and physical activity level.**

| Variables and Categories | Total (n) | % (95% CI)* |
|---|---|---|
| Age Range | | |
| 60–69 | 573 | 57.2 (53.0–61.5) |
| ≥70 | 446 | 42.8 (38.5–47.0) |
| Gender | | |
| Female | 632 | 59.7 (56.9–62.4) |
| Male | 387 | 40.3 (37.6–43.1) |
| Race/skin color [a] | | |
| White | 611 | 64.9 (60.1–69.7) |
| Non-white | 400 | 35.1 (30.3–39.9) |
| Marital status [b] | | |
| With a partner | 515 | 52.8 (45.6–57.0) |
| Without a partner | 500 | 47.2 (43.0–51.4) |
| Education [c] | | |
| Never been to school | 90 | 6.9 (5.0–8.7) |
| Elementary school | 458 | 40.7 (36.2–45.3) |
| Junior high school | 153 | 15.1 (12.7–17.4) |
| High school | 172 | 17.9 (15.4–20.3) |
| College | 140 | 19.54–14.8–24.2) |
| Occupation or Volunteer work [d] | | |
| No | 745 | 70.4 (66.7–74.0) |
| Yes | 272 | 29.6 (26.0–33.3) |
| Per capita income | | |
| ≤ 0.5 | 350 | 35.3 (29.4–41.2) |
| 0.5–1.0 | 200 | 16.6 (13.2–20.1) |
| 1 to 2.5 | 289 | 28.2 (23.2–33.2) |
| ≥2.5 | 180 | 19.9 (16.6–23.2) |
| **CHC number** | | |
| 0–1 | 338 | 32.3 (28.9–35.7) |
| 2 | 192 | 19.2 (16.6–21.7) |
| 3 | 159 | 16.6 (13.9–19.2) |
| 4 | 138 | 12.9 (10.5–15.4) |
| ≥5 | 192 | 19.0 (16.0–22.0) |
| Multimorbidity | | |
| No | 338 | 32.3 (28.9–35.7) |
| Yes | 681 | 67.7 (64.3–71.1) |
| Physical activity level [e] | | |
| Insufficient | 719 | 69.9 (65.8–73.9) |
| Sufficient | 298 | 30.1 (26.1–34.2) |

*Weighted percentage; CHC–Chronic Health Conditions; 95% CI– 95% confidence interval

[a] 8 missing

[b] 4 missing

[c] 6 missing

[d] 2 missing

[e] 2 missing.

**Table 2. Distribution and regression analysis between physical activity level and number of chronic health conditions.**

| Variables and Categories | Physical activity level | | | | | P-value* | Crude PR (95% CI) | P-value | Adjusted PR** (95% CI) | P-value |
|---|---|---|---|---|---|---|---|---|---|---|
| | Insufficient | | Sufficient | | | | | | | |
| | n | % (95% CI) | n | % (95% CI) | | | | | | |
| CHC number | | | | | | 0.310 | | | | |
| 0–1 | 243 | 73.2 (67.7–78.1) | 95 | 26.8 (21.9–32.3) | | | 1 | | 1 | |
| 2 | 126 | 63.7 (54.6–72.0) | 66 | 36.3 (28.0–45.4) | | | 1.35 (0.99–1.84) | 0.055 | **1.38 (1.02–1.88)** | **0.039** |
| 3 | 116 | 72.4 (62.7–80.3) | 42 | 27.6 (19.7–37.3) | | | 1.03 (0.72–1.48) | 0.874 | 1.05 (0.74–1.49) | 0.799 |
| 4 | 95 | 66.3 (56.9–74.5) | 43 | 33.7 (25.5–43.1) | | | 1.26 (0.91–1.73) | 0.158 | **1.37 (1.00–1.87)** | **0.049** |
| ≥5 | 139 | 70.7 (61.2–78.7) | 52 | 29.3 (21.3–38.8) | | | 1.09 (0.77–1.56) | 0.619 | 1.25 (0.89–1.76) | 0.194 |

Weighted percentage; CHC–Chronic Health Conditions; PR–Prevalence ratio; 95% CI– 95% confidence interval

* chi-squared test

** adjusted by age range, education and gender.

prevalence of a sufficient level of physical activity was observed among older adults with two and four chronic conditions, while physical activity maintenance was observed among those with three and five or more chronic conditions. Older adults with multimorbidity who were 70 years or older, female, without a partner, and had a per capita income of 1 to 2.5 were more likely to achieve a sufficient level of physical activity compared to their peers without multimorbidity.

The prevalence of multimorbidity among older adults found in the present study was 67.7%. This result is coherent with a previous study conducted in the city of São Paulo [45], though its result is above the international [19, 22, 46] and the national average [6, 9, 13, 47, 48] among other studies.

It is essential to consider methodological differences when observing the divergence of prevalence in some studies, such as the number of chronic conditions in the definitions of multimorbidity. Studies may adopt broader or more restrictive criteria for identifying diseases, which will result in different prevalences [49]. Nevertheless, it is known that with the increase in life expectancy, there is an increase in the prevalence of chronic diseases [6, 9, 10, 50] and the growth in the number of older adults with multimorbidity is worrisome and needs to be analyzed by considering the importance of facilitating their access to health care, costs, quality of care, as well as the imminent risks of worsening pathological processes.

Practicing physical activity reduces the risk of mortality from chronic diseases [20], with a consensus in the literature to encourage the maintenance of physical activity among older adults who live with chronic health conditions. In this study, older adults with two chronic health conditions have a 38% higher likelihood, while those with and four chronic health conditions have a 37% higher likelihood of reaching a sufficient level of physical activity when compared to those with up to one chronic health condition. In addition, the of older adults with three and five or more chronic conditions was maintained with sufficient level of physical activity. Such results partially agree with the national literature [27, 51] though it shows inconsistency with international results [14, 34, 52–54]. The study carried out by Christofoletti et al. [27] with information from the Risk and Protective Factors Surveillance System for Chronic Diseases, did not identify differences between the prevalence of physical activity (leisure and commuting) among people with 60 years or older who live with two, three, or four or more chronic health condition when compared to those with up to one condition health chronic. On the other hand, in a Health Survey carried out in Spain and analyzed by Cimarras-Otal et al. [52], negative evidence of an increase in chronic health condition and physical activity for both males and females aged 74 years or older was observed.

**Table 3. Distribution of sufficient level of physical activity among older adults with and without multimorbidity according to socioeconomic, demographic characteristics and regression analysis.**

| Variables and Categories | without multimorbidity* | | with multimorbidity | | Crude PR (95% CI) | P-value | Adjusted PR** (95% CI) | P-value |
|---|---|---|---|---|---|---|---|---|
| | Physical activity level | | | | | | | |
| | sufficient | | sufficient | | | | | |
| | n | % (95% CI) | n | % (95% CI) | | | | |
| Age Range | | | | | | | | |
| 60–69 | 69 | 32.9 (26.7–39.8) | 126 | 34.7 (28.3–41.9) | 1.05 (0.81–1.37) | 0.689 | 1.07 (0.83–1.39) | 0.582 |
| ≥70 | 26 | 17.6 (11.6–25.8) | 77 | 27.9 (22.4–34.2) | **1.59 (1.00–2.53)** | **0.049** | **1.77 (1.13–2.77)** | **0.013** |
| Gender | | | | | | | | |
| Female | 32 | 18.0 (13.2–24.1) | 131 | 28.9 (23.4–35.2) | **1.61 (1.12–2.32)** | **0.011** | **1.65 (1.16–2.36)** | **0.006** |
| Male | 63 | 34.0 (27.2–41.6) | 72 | 37.3 (29.5–45.9) | 1.10 (0.80–1.51) | 0.570 | 1.05 (0.76–1.46) | 0.748 |
| Race/skin color | | | | | | | | |
| White | 54 | 24.5 (18.4–32.0) | 117 | 31.6 (25.8–37.9) | 1.29 (0.91–1.82) | 0.154 | 1.32 (0.94–1.83) | 0.104 |
| Non-white | 39 | 30.6 (23.1–39.4) | 86 | 32.3 (25.8–39.5) | 1.05 (0.75–1.47) | 0.762 | 1.15 (0.80–1.63) | 0.450 |
| Marital status | | | | | | | | |
| With a partner | 55 | 29.4 (22.4–37.4) | 107 | 33.8 (27.6–40.5) | 1.15 (0.83–1.58) | 0.393 | 1.16 (0.84–1.60) | 0.367 |
| Without a partner | 39 | 23.4 (17.4–30.6) | 96 | 29.7 (23.7–36.5) | 1.27 (0.90–1.79) | 0.166 | **1.43 (1.03–1.99)** | **0.032** |
| Education (years) | | | | | | | | |
| Never been to school | 4 | 11.6 (4.0–29.0) | 14 | 18.6 (11.1–29.5) | 1.61 (0.54–4.82) | 0.387 | 2.17 (0.75–6.25) | 0.148 |
| Elementary school | 38 | 24.3 (17.4–32.9) | 80 | 25.8 (21.1–31.2) | 1.06 (0.74–1.53) | 0.739 | 1.28 (0.89–1.85) | 0.179 |
| Junior high school | 16 | 25.3 (15.3–38.8) | 26 | 33.0 (20.8–48.1) | 1.31 (0.67–2.55) | 0.430 | 1.30 (0.66–2.54) | 0.446 |
| High school | 20 | 31.1 (20.6–44.1) | 38 | 37.2 (26.2–49.8) | 1.20 (0.78–1.84) | 0.410 | 1.24 (0.77–1.98) | 0.369 |
| College | 16 | 35.5 (22.6–50.9) | 45 | 42.9 (33.5–52.8) | 1.21 (0.75–1.94) | 0.433 | 1.23 (0.79–1.94) | 0.352 |
| Occupation or Volunteer work | | | | | | | | |
| No | 63 | 25.3 (19.5–32.1) | 143 | 28.7 (24.5–33.4) | 1.14 (0.84–1.53) | 0.398 | 1.28 (0.98–1.67) | 0.075 |
| Yes | 32 | 29.7 (20.6–40.8) | 60 | 40.2 (29.9–51.3) | 1.35 (0.89–2.06) | 0.160 | 1.32 (0.85–2.04) | 0.210 |
| Per capita income | | | | | | | | |
| ≤ 0.5 | 35 | 25.2 (17.8–34.5) | 62 | 32.4 (24.4–41.6) | 1.28 (0.88–1.88) | 0.195 | 1.35 (0.92–1.99) | 0.126 |
| 0.5–1.0 | 20 | 29.6 (19.7–41.8) | 34 | 23.1 (16.0–32.3) | 0.78 (0.48–1.27) | 0.319 | 0.93 (0.57–1.51) | 0.770 |
| 1 to 2.5 | 20 | 18.5 (10.9–29.6) | 54 | 29.1 (21.7–37.7) | 1.57 (0.84–2.94) | 0.152 | **1.83 (1.00–3.33)** | **0.047** |
| ≥2.5 | 20 | 44.4 (28.4–61.7) | 53 | 40.8 (30.6–51.9) | 0.92 (0.57–1.47) | 0.723 | 0.97 (0.61–1.53) | 0.887 |

Weighted percentage; PR–Prevalence ratio; 95% CI– 95% confidence interval

*Reference category

**adjusted by age range, education and gender.

When analyzing the results of the association between physical activity and the number of chronic diseases, it is essential to consider the burden of diseases and their severity, as well as the degree of control over them. Additionally, rehabilitation programs that emphasize physical activity can influence this relationship. It is possible that, more than the quantity of diseases, the specific type of disease and whether it is effectively controlled or not, play critical roles in the relationship between multimorbidity and physical activity. The different methods used to identify and explore the physical activity level in each of the above-mentioned studies may explain their divergences with the present study. For example, Gomes et al. [51] measured physical activity through the long version of the IPAQ, using the domains of leisure and commuting as one, despite the similarity of the methodology used in that study, the authors chose to stratify by gender. On the other hand, the study by Christofoletti et al. [27] used a keyboard based on information from the World Health Organization and chose to analyze each domain

of physical activity (leisure, occupational, commuting, domestic and total). In contrast with these surveys, international studies retrieved IPAQ information and used domains in different ways. Other causes underlying the evidenced differences may be related to the division of the sample into narrower age groups and configurations with other variables, which could reveal other associations.

The study evidenced the fact that older adults with multimorbidity and who are 70 years or older, who do not live with a partner, and having a per capita income of 1 to 2.5, demonstrated higher likelihood of achieving a sufficient level of physical activity when compared to their peers without multimorbidity. Up to this moment, no evidence has been found in the literature that corroborates these findings. This suggests that additional research is needed to explore and validate these findings, as well as to explore strategies to focus on increasing physical activity among these groups. These results may be attributed to changes or strengthening of healthy habits with the accumulation of chronic health conditions [30]. Another possible explanation for this association may be the possibility of reverse causality, which is when the occurrence of the event influences the occurrence of exposure, as this can occur in cross-sectional studies, such as health surveys.

A positive association between achieving a sufficient level of physical activity was evidenced among women with multimorbidity when compared to those without multimorbidity. In this context, even females with ≤1 chronic health condition failed to reach the sufficient level of physical activity, an important level to meet the needs of preventing, controlling or even reducing the probability of developing multimorbidity, bearing the development of actions directed to the group identified in this study.

Considering the use of the long version of the IPAQ and with the domains of physical activity (leisure and commuting), the study by Gomes et al. [51] identified, among females, a greater probability of reaching insufficient levels of physical activity with the increase in the number of chronic health condition. Such results do not corroborate with the present study. Thus, we emphasize the need to deepen the behavior of physical activity practice among individuals with multimorbidity according to gender, an issue that has little evidence in the literature.

The choice to use domains such as leisure and commuting in the present study was due to their greater relevance to address public health policies and programs [55], and to encourage the practice of walking, cycling, water aerobics and gymnastics, which are activities recommended by international [33] and national [32] guidelines for older adults' groups. It is known that walking is the main modality practiced in Brazil [56], it can be stimulated both for leisure and commuting. It is a low-impact activity, requiring sites with less complexity and lower costs compared to other activities.

Some restrictions must be considered regarding this study. Its cross-sectional design prevents the identification of the direction of the association. Another aspect to be mentioned when interpreting its results is the fact that no laboratorial and/or clinical tests were used to assess or measure the level of physical activity and health conditions, as these procedures are costly and time-consuming. This fact may have contributed to overestimated results. Our measure of multimorbidity was limited toward several chronic health condition. All the diseases were given equal weight for the analysis without any assessment of their severity. However, the method for measuring the practice of physical activity has strengths, such as the use of the controlled version of the IPAQ, a validated instrument and one of the most used in the world, which allows independent assessment of the different domains of physical activity (leisure, home, work and commuting) or putting some domains together, to bear comparisons with other studies, good acceptability of the participants and the remembrance of a week before the interview, which allows interviewees to ease their memories of the practice. The present findings can only be extrapolated to older adults residents of the city of São Paulo, Brazil.

This study identified relevant aspects of multimorbidity and physical activity level in older adults living in the Brazil largest city. A high prevalence of multimorbidity and a low frequency of level of sufficient physical activity were identified, suggesting the need to reinforce monitoring of physical activity and encourage the promotion of physical activity among older adults who face health challenges or not. In addition, older adults with two and four chronic conditions were more likely to achieve a reaching a sufficient level of physical activity, while physical activity maintenance was observed among those with three and five or more chronic conditions. This emphasizes the need to strengthen public health measures aimed at encouraging physical activity. Older adults with multimorbidity in the highest age group, female, who did not have a partner, and with a per capita income of 1 to 2.5 were more likely to achieve sufficient levels of physical activity when compared to their peers with even one chronic health condition. This suggests that sociodemographic aspects should be considered in the planning of public policies addressed to the practice of physical activity among people with and without multimorbidity.

## Supporting information

**S1 File. Minimal data set.** Set of data analysed for each individual include in this study. (XLSX)

## Acknowledgments

Municipal Health Department of the City of São Paulo.

## Author Contributions

**Conceptualization:** Bruno Holanda Ferreira, Ricardo Goes de Aguiar, Edige Felipe de Sousa Santos, Chester Luiz Galvão Cesar, Moisés Goldbaum, Camila Nascimento Monteiro.

**Data curation:** Bruno Holanda Ferreira, Chester Luiz Galvão Cesar, Moisés Goldbaum, Camila Nascimento Monteiro.

**Formal analysis:** Bruno Holanda Ferreira, Ricardo Goes de Aguiar, Edige Felipe de Sousa Santos, Chester Luiz Galvão Cesar, Moisés Goldbaum, Camila Nascimento Monteiro.

**Investigation:** Bruno Holanda Ferreira, Edige Felipe de Sousa Santos, Chester Luiz Galvão Cesar, Moisés Goldbaum, Camila Nascimento Monteiro.

**Methodology:** Bruno Holanda Ferreira, Ricardo Goes de Aguiar, Chester Luiz Galvão Cesar, Moisés Goldbaum, Camila Nascimento Monteiro.

**Project administration:** Moisés Goldbaum.

**Supervision:** Bruno Holanda Ferreira, Edige Felipe de Sousa Santos, Chester Luiz Galvão Cesar, Moisés Goldbaum, Camila Nascimento Monteiro.

**Validation:** Bruno Holanda Ferreira, Moisés Goldbaum, Camila Nascimento Monteiro.

**Visualization:** Bruno Holanda Ferreira, Moisés Goldbaum.

**Writing – original draft:** Bruno Holanda Ferreira, Ricardo Goes de Aguiar, Edige Felipe de Sousa Santos, Chester Luiz Galvão Cesar, Moisés Goldbaum, Camila Nascimento Monteiro.

**Writing – review & editing:** Bruno Holanda Ferreira, Ricardo Goes de Aguiar, Edige Felipe de Sousa Santos, Chester Luiz Galvão Cesar, Moisés Goldbaum, Camila Nascimento Monteiro.

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
