## [Decision Letter · Decision Letter 0]

20 Sep 2023

PONE-D-23-19647Physical activity among older adults with multimorbidity: Evidence from a population-based health surveyPLOS ONE

Dear Dr. Ferreira,

Thank you for submitting your manuscript to PLOS ONE. After careful consideration, we feel that it has merit but does not fully meet PLOS ONE’s publication criteria as it currently stands. Therefore, we invite you to submit a revised version of the manuscript that addresses the points raised during the review process.

Below are comments and suggestions made by two different reviewers.

I request that the questions indicated be addressed, especially those that refer to the major issues highlighted by reviewer 1.==============================

We look forward to receiving your revised manuscript.

Kind regards,

Mathias Roberto Loch, Ph.D

Academic Editor

PLOS ONE

Additional Editor Comments:

Reviewer 1

Dear Authors,

Congratulations on the writing of the article "physical activity among older adults with multimorbidity: Evidence from a population-based health survey," submitted to the journal PLOS One.

The study aims to estimate the prevalence of multimorbidity and physical activity, as well as to analyze the relationship between meeting recommended levels of physical activity and multimorbidity, according to the characteristics of older adults residents in São Paulo, Brazil.

Major Concerns:

The study's introduction shows important information, but it does not clearly state the research problem. The authors argue that there is a relationship between physical activity and morbidities, implying that more older adults more active (those who meet recommendations) have a lower chance of having multimorbidity. Or is the hypothesis that older adults with morbidities are more active? The authors need to clarify this hypothesis. Additionally, they should present results from studies that have already tested this relationship to demonstrate the knowledge gap they are addressing. The fact that the study is conducted in a high-middle-income country (Brazil) cannot be the sole justification for the research.

In the methods section, the authors have chosen to perform an association analysis with the variable multimorbidity (two or more morbidities), but this information is presented categorically according to the number of morbidities. These results show that only having two morbidities was associated with the prevalence of being active, suggesting that among elderly individuals with morbidities, the prevalence of being active is higher than in those without or with one morbidity. Would this result be expected? The authors do not explain why individuals with four or more morbidities, for example, do not have a lower prevalence of being active. Wouldn't it be expected that individuals with more morbidities would have a lower prevalence of being active? This is not clear in the article.

Minor Concerns:

Page 2, line 38 – There is no need to present the classification of insufficient level of physical activity.

Page 2, lines 43 and 44 – Adjust the text to "two chronic diseases" and "three or more chronic diseases” no use 2 or 3.

Page 3, line 77 – The authors mention that one of the factors contributing to the increase in the prevalence of insufficient levels of physical activity is the transition from a professional life to retirement. However, there are many other factors, and this is just one of them. If other factors are not described, I suggest excluding this argument, as it is not mentioned further in the text.

Page 3, line 78 – The information about the number of deaths attributed to physical inactivity is out of place. This information could be placed where problems related to physical inactivity are discussed.

Page 4, line 85 – The authors mention that in the study by Zho et al., both regular and irregular physical activity were associated with chronic diseases. The text is confusing; this is in comparison to those who are inactive. The authors can clarify this point.

Page 4, line 90 – The study's focus is on physical activity, so it is suggested to avoid including sedentary behavior. This is another outcome that is not the focus of the study and providing this information might confuse the readers.

Method:

Page 5, line 114 – Mention that the data were collected before the COVID-19 pandemic.

Results:

Page 8, Table 1 – Display the unit of measurement for per capita income.

Page 8, line 200 – The authors could test the relationship between physical activity and multimorbidity, as this was the justification outlined in the introduction.

Page 8, Table 2 – Standardize the number of decimal places in the table (e.g., 63.7 (54.6-72.0)).

Page 9, Table 3 – The authors should consider that the group aged 80 has only six inactive elderly individuals without multimorbidity, which is a very low proportion. How might this impact the results? Could the existing association be due to the fact that almost all elderly individuals over 80 have multimorbidity?

Discussion:

Page 10, line 217 – The authors state that physical activity was maintained for individuals with three or more chronic conditions. This result is not presented.

Page 10, line 230 – One of the main study results shows that elderly individuals with two chronic conditions have a 37% higher prevalence of sufficient levels of physical activity. There is no justification for the presented results. Methodological differences are important, but beyond them, what explanations could account for this relationship?

Page 12, line 298 – The authors need to explore how the results can be applied in the field of public health. How can the results be considered by public policy planners? It is also suggested to remove the reduction of sedentary behavior, as this behavior was not evaluated in the study.

Reviewer 2

General comments

Congratulations to the authors for the well conduction of this study and construction of the present manuscript. This article aimed to estimate the prevalence of multimorbidity and physical activity, as well as analyze the relationship between a sufficient level of physical activity and multimorbidity, considering sociodemographic characteristics of residents in São Paulo, Brazil. The study provides results on a context that is still little explored in the elderly population, especially those residing in developing countries. Therefore, the study may contribute to increasing knowledge about the relationship between multimorbidities and the level of habitual physical activity of elderly people with different sociodemographic characteristics and comparing these results with elderly people living in other realities (countries). In general, the article is well structured and long, with the strong point being a representative sample of elderly people from one of the most populous municipalities in the world. Below I present my specific comments and some suggestions for modifications.

Specific comments

Abstract:

Comment 1- Page 2, line 34, include in the objective that the study was carried out with older people.

Introduction:

Comment 2- Page 4, lines 89-95. The sentence is too long and compromises the understanding of the message to be transmitted. The justification needs to be more forceful, as the authors justify only the fact that the analyzed context is little explored in low- and middle-income countries like Brazil. Also present other elements that strengthen the justification of the study.

Methodology:

Comment 3- Page 4, line 99. Replace the subtitle Methods with Materials and Methods, following the Journal's guidelines.

Comment 4- Present in more detail what ISA-Capital 2015 is or a reference that better explains what this project/program is.

Comment 5- Page 4, line 103. Include reference to this population data.

Discussion

Comment 6- Some results of this study are intriguing and should be further discussed. I highlight three main aspects below:

6.1) Why is the prevalence of multimorbidity in the elderly in the city of São Paulo higher than the national and international average? What are the possible reasons for this result? This inference must be presented on page 10, between lines 221 and 227.

6.2) Elderly people with multimorbidity are more likely to be sufficiently active compared to those with < 2 diseases. An important point that may have influenced the results of the study is the failure to control the burden of diseases, their severity and their control. Perhaps more important than the number of comorbidities is the type of disease and whether it is controlled or not. I missed the discussion on this aspect. The authors only mentioned this point as a limitation of the study.

6.3) The study showed that elderly people with multimorbidity and aged 80 or over, who do not live with a partner, were more likely to achieve a sufficient level of physical activity when compared to their peers without multimorbidity. How can these results be explained? Are these results consistent with other studies? Attributing this result solely to reverse causality is too simplistic.

Reviewers' comments:

Reviewer's Responses to Questions

**Comments to the Author**

1. Is the manuscript technically sound, and do the data support the conclusions?

Reviewer #1: Yes

Reviewer #2: Partly

2. Has the statistical analysis been performed appropriately and rigorously? 

Reviewer #1: Yes

Reviewer #2: Yes

3. Have the authors made all data underlying the findings in their manuscript fully available?

Reviewer #1: Yes

Reviewer #2: No

4. Is the manuscript presented in an intelligible fashion and written in standard English?

Reviewer #1: Yes

Reviewer #2: Yes

5. Review Comments to the Author

Reviewer #1: Dear Authors,

Congratulations on the writing of the article "physical activity among older adults with multimorbidity: Evidence from a population-based health survey," submitted to the journal PLOS One.

The study aims to estimate the prevalence of multimorbidity and physical activity, as well as to analyze the relationship between meeting recommended levels of physical activity and multimorbidity, according to the characteristics of older adults residents in São Paulo, Brazil.

Major Concerns:

The study's introduction shows important information, but it does not clearly state the research problem. The authors argue that there is a relationship between physical activity and morbidities, implying that more older adults more active (those who meet recommendations) have a lower chance of having multimorbidity. Or is the hypothesis that older adults with morbidities are more active? The authors need to clarify this hypothesis. Additionally, they should present results from studies that have already tested this relationship to demonstrate the knowledge gap they are addressing. The fact that the study is conducted in a high-middle-income country (Brazil) cannot be the sole justification for the research.

In the methods section, the authors have chosen to perform an association analysis with the variable multimorbidity (two or more morbidities), but this information is presented categorically according to the number of morbidities. These results show that only having two morbidities was associated with the prevalence of being active, suggesting that among elderly individuals with morbidities, the prevalence of being active is higher than in those without or with one morbidity. Would this result be expected? The authors do not explain why individuals with four or more morbidities, for example, do not have a lower prevalence of being active. Wouldn't it be expected that individuals with more morbidities would have a lower prevalence of being active? This is not clear in the article.

Minor Concerns:

Page 2, line 38 – There is no need to present the classification of insufficient level of physical activity.

Page 2, lines 43 and 44 – Adjust the text to "two chronic diseases" and "three or more chronic diseases” no use 2 or 3.

Page 3, line 77 – The authors mention that one of the factors contributing to the increase in the prevalence of insufficient levels of physical activity is the transition from a professional life to retirement. However, there are many other factors, and this is just one of them. If other factors are not described, I suggest excluding this argument, as it is not mentioned further in the text.

Page 3, line 78 – The information about the number of deaths attributed to physical inactivity is out of place. This information could be placed where problems related to physical inactivity are discussed.

Page 4, line 85 – The authors mention that in the study by Zho et al., both regular and irregular physical activity were associated with chronic diseases. The text is confusing; this is in comparison to those who are inactive. The authors can clarify this point.

Page 4, line 90 – The study's focus is on physical activity, so it is suggested to avoid including sedentary behavior. This is another outcome that is not the focus of the study and providing this information might confuse the readers.

Method:

Page 5, line 114 – Mention that the data were collected before the COVID-19 pandemic.

Results:

Page 8, Table 1 – Display the unit of measurement for per capita income.

Page 8, line 200 – The authors could test the relationship between physical activity and multimorbidity, as this was the justification outlined in the introduction.

Page 8, Table 2 – Standardize the number of decimal places in the table (e.g., 63.7 (54.6-72.0)).

Page 9, Table 3 – The authors should consider that the group aged 80 has only six inactive elderly individuals without multimorbidity, which is a very low proportion. How might this impact the results? Could the existing association be due to the fact that almost all elderly individuals over 80 have multimorbidity?

Discussion:

Page 10, line 217 – The authors state that physical activity was maintained for individuals with three or more chronic conditions. This result is not presented.

Page 10, line 230 – One of the main study results shows that elderly individuals with two chronic conditions have a 37% higher prevalence of sufficient levels of physical activity. There is no justification for the presented results. Methodological differences are important, but beyond them, what explanations could account for this relationship?

Page 12, line 298 – The authors need to explore how the results can be applied in the field of public health. How can the results be considered by public policy planners? It is also suggested to remove the reduction of sedentary behavior, as this behavior was not evaluated in the study.

Reviewer #2: General comments

Congratulations to the authors for the well conduction of this study and construction of the present manuscript. This article aimed to estimate the prevalence of multimorbidity and physical activity, as well as analyze the relationship between a sufficient level of physical activity and multimorbidity, considering sociodemographic characteristics of residents in São Paulo, Brazil. The study provides results on a context that is still little explored in the elderly population, especially those residing in developing countries. Therefore, the study may contribute to increasing knowledge about the relationship between multimorbidities and the level of habitual physical activity of elderly people with different sociodemographic characteristics and comparing these results with elderly people living in other realities (countries). In general, the article is well structured and long, with the strong point being a representative sample of elderly people from one of the most populous municipalities in the world. Below I present my specific comments and some suggestions for modifications.

Specific comments

Abstract:

Comment 1- Page 2, line 34, include in the objective that the study was carried out with older people.

Introduction:

Comment 2- Page 4, lines 89-95. The sentence is too long and compromises the understanding of the message to be transmitted. The justification needs to be more forceful, as the authors justify only the fact that the analyzed context is little explored in low- and middle-income countries like Brazil. Also present other elements that strengthen the justification of the study.

Methodology:

Comment 3- Page 4, line 99. Replace the subtitle Methods with Materials and Methods, following the Journal's guidelines.

Comment 4- Present in more detail what ISA-Capital 2015 is or a reference that better explains what this project/program is.

Comment 5- Page 4, line 103. Include reference to this population data.

Discussion

Comment 6- Some results of this study are intriguing and should be further discussed. I highlight three main aspects below:

6.1) Why is the prevalence of multimorbidity in the elderly in the city of São Paulo higher than the national and international average? What are the possible reasons for this result? This inference must be presented on page 10, between lines 221 and 227.

6.2) Elderly people with multimorbidity are more likely to be sufficiently active compared to those with < 2 diseases. An important point that may have influenced the results of the study is the failure to control the burden of diseases, their severity and their control. Perhaps more important than the number of comorbidities is the type of disease and whether it is controlled or not. I missed the discussion on this aspect. The authors only mentioned this point as a limitation of the study.

6.3) The study showed that elderly people with multimorbidity and aged 80 or over, who do not live with a partner, were more likely to achieve a sufficient level of physical activity when compared to their peers without multimorbidity. How can these results be explained? Are these results consistent with other studies? Attributing this result solely to reverse causality is too simplistic.

6. PLOS authors have the option to publish the peer review history of their article (what does this mean?). If published, this will include your full peer review and any attached files.

Reviewer #1: No

Reviewer #2: No

---

## [Author Response · Author response to Decision Letter 0]

4 Nov 2023

São Paulo, 

Letter in response to the editor

Dear PLOS ONE Editor,

We would like to thank the reviewers for their careful review and feedback on the article “Physical activity among older adults with multimorbidity: Evidence from a population-based health survey” and for providing valuable suggestions to improve it. We acknowledge that the recommendations made by the reviewers have helped us to improve the quality of our article. The authors’ clarifications and justifications are presented below.

Answers to the Editor’s specific comments:

• As requested, revisions have been made to conform to Plos One style.

• The cover letter was revised as requested and it was indicated that there are no restrictions on the minimum data set. 

• Below are the comments made for each signal provided by the reviewers.

Reviewer #1

Major Concerns:

INTRODUCTION

1) The study's introduction shows important information, but it does not clearly state the research problem. The authors argue that there is a relationship between physical activity and morbidities, implying that more older adults more active (those who meet recommendations) have a lower chance of having multimorbidity. Or is the hypothesis that older adults with morbidities are more active? The authors need to clarify this hypothesis. Additionally, they should present results from studies that have already tested this relationship to demonstrate the knowledge gap they are addressing. The fact that the study is conducted in a high-middle-income country (Brazil) cannot be the sole justification for the research.

Answers: Adjustments were made to better understand the paragraph on changes in behavior after diagnosis of chronic disease. Page 04, lines 81-97.

Information was included to better understand the research problem. Page 4, lines 93-102.

METHODS SECTION

2) In the methods section, the authors have chosen to perform an association analysis with the variable multimorbidity (two or more morbidities), but this information is presented categorically according to the number of morbidities. These results show that only having two morbidities was associated with the prevalence of being active, suggesting that among elderly individuals with morbidities, the prevalence of being active is higher than in those without or with one morbidity. Would this result be expected? The authors do not explain why individuals with four or more morbidities, for example, do not have a lower prevalence of being active. Wouldn't it be expected that individuals with more morbidities would have a lower prevalence of being active? This is not clear in the article.

Answers: The information has been added. Page 11-12, lines 262-267.

MINOR CONCERNS:

3) Page 2, line 38 – There is no need to present the classification of insufficient level of physical activity.

Answers: Adjustments were made to the text, excluding the classification of insufficiently active individuals. Page 2, line 37-38.

4) Page 2, lines 43 and 44 – Adjust the text to "two chronic diseases" and "three or more chronic diseases” no use 2 or 3.

Answers: Adjustments were made to the text. Page 2, lines 42-43.

5) Page 3, line 77 – The authors mention that one of the factors contributing to the increase in the prevalence of insufficient levels of physical activity is the transition from a professional life to retirement. However, there are many other factors, and this is just one of them. If other factors are not described, I suggest excluding this argument, as it is not mentioned further in the text.

Answers: According to the suggestion, the factor that can indicate insufficiently active physical activity was removed from the text. Page 3, lines 76-77.

6) Page 3, line 78 – The information about the number of deaths attributed to physical inactivity is out of place. This information could be placed where problems related to physical inactivity are discussed.

Answers: By removing one of the factors that leads to insufficient physical activity (Page 3, lines 77-78), information about deaths was included in the problems attributed to physical inactivity. Page 3, lines 77-78.

7) Page 4, line 85 – The authors mention that in the study by Zho et al., both regular and irregular physical activity were associated with chronic diseases. The text is confusing; this is in comparison to those who are inactive. The authors can clarify this point.

Answers: Adjustments were made to the text to correct the information. Page 4, lines 84-86.

8) Page 4, line 90 – The study's focus is on physical activity, so it is suggested to avoid including sedentary behavior. This is another outcome that is not the focus of the study and providing this information might confuse the readers.

Answers: Adjustments were made to the text, excluding the term “sedentary behavior”. Page 4, lines 90.

METHOD:

9) Page 5, line 114 – Mention that the data were collected before the COVID-19 pandemic.

Answers: An adjustment was made to the text to inform that the data was collected before the COVID-19 pandemic. Page 5, line 123.

RESULTS:

10) Page 8, Table 1 – Display the unit of measurement for per capita income.

Answers: The adjustment has been added. Page 8 Table 1.

11) Page 8, line 200 – The authors could test the relationship between physical activity and multimorbidity, as this was the justification outlined in the introduction.

Answers – The chi-square test was included in the statistical treatment. Page 7, lines 180-181. In addition, the test result (p-value) was included in table 2, Page 9.

12) Page 8, Table 2 – Standardize the number of decimal places in the table (e.g., 63.7 (54.6-72.0)).

Answers: Adjustments were made to the 95%CI for the category of two chronic diseases. Page 9, Table 2.

13) Page 9, Table 3 – The authors should consider that the group aged 80 has only six inactive elderly individuals without multimorbidity, which is a very low proportion. How might this impact the results? Could the existing association be due to the fact that almost all elderly individuals over 80 have multimorbidity?

Answers: The age range was modified to two categories (60-69 and ≥70) in the Materials and Methods (Page 6, line 157), Table 1 and 3 (page 8 and 10). This made it possible to increase the number of individuals between categories and improve the measure of effect. Moreover, the Poisson regression (adjusted) was reconducted to accommodate the new age range, leading to modifications in certain values.

DISCUSSION:

14) Page 10, line 217 – The authors state that physical activity was maintained for individuals with three or more chronic conditions. This result is not presented.

Answers: The information about the maintenance of the prevalence of sufficiently active individuals was added to the text. Page 10, lines 231-233.

15) Page 10, line 230 – One of the main study results shows that elderly individuals with two chronic conditions have a 37% higher prevalence of sufficient levels of physical activity. There is no justification for the presented results. Methodological differences are important, but beyond them, what explanations could account for this relationship?

Answers: Information was added. Page 11, lines 239-242 / 250-253.

16) Page 12, line 298 – The authors need to explore how the results can be applied in the field of public health. How can the results be considered by public policy planners? It is also suggested to remove the reduction of sedentary behavior, as this behavior was not evaluated in the study.

Answers: Information was added for the field of public health and public policy planners. The information is available on pages 13 and 14, lines 324-326 and 333-335.

Sedentary behavior was removed. The information is available on page 13, lines 327.

Reviewer #2

ABSTRACT:

1) Comment 1- Page 2, line 34, include in the objective that the study was carried out with older people.

Answers: The sample of older adults was added to the objective written in the abstract. The information is available on page 2, line 31.

INTRODUCTION

2) Comment 2- Page 4, lines 89-95. The sentence is too long and compromises the understanding of the message to be transmitted. The justification needs to be more forceful, as the authors justify only the fact that the analyzed context is little explored in low- and middle-income countries like Brazil. Also present other elements that strengthen the justification of the study.

Answers: Information was included for a better understanding of the research problem. The information is available on page 4, lines 98-102.

METHODOLOGY

3) Page 4, line 99. Replace the subtitle Methods with Materials and Methods, following the Journal's guidelines.

Answers: The subtitle Materials and Methods was added. The information is available on page 4, line 106.

4) Present in more detail what ISA-Capital 2015 is or a reference that better explains what this project/program is.

Answers: The reference of the ISA-Capital project was added. The information is available on page 4, line 114.

5) Page 4, line 103. Include reference to this population data.

Answers: The population data of the city of São Paulo were updated and added to the reference. The information is available on page 4, lines 107-111.

DISCUSSION

6) Some results of this study are intriguing and should be further discussed. I highlight three main aspects below:

6.1) Why is the prevalence of multimorbidity in the elderly in the city of São Paulo higher than the national and international average? What are the possible reasons for this result? This inference must be presented on page 10, between lines 221 and 227.

Answers: Information that can explain differences in the prevalence of multimorbidity was added. The information is available on page 11, lines 239-242.

6.2) Elderly people with multimorbidity are more likely to be sufficiently active compared to those with < 2 diseases. An important point that may have influenced the results of the study is the failure to control the burden of diseases, their severity and their control. Perhaps more important than the number of comorbidities is the type of disease and whether it is controlled or not. I missed the discussion on this aspect. The authors only mentioned this point as a limitation of the study.

Answers: The suggested information was added. The information is available on page 11-12, lines 262-267.

6.3) The study showed that elderly people with multimorbidity and aged 80 or over, who do not live with a partner, were more likely to achieve a sufficient level of physical activity when compared to their peers without multimorbidity. How can these results be explained? Are these results consistent with other studies? Attributing this result solely to reverse causality is too simplistic.

Answers: 6.3) The age range was modified to two categories (60-69 and ≥70) in the Materials and Methods (Page 6, line 157), Table 1 and 3 (page 8 and 10).

Adjustments were made to the paragraphs to suit the request. The information is available on page 12, lines 280-285.

Attentiously,

Bruno Holanda Ferreira; Ricardo Goes de Aguiar, Edige Felipe de Sousa Santos, Chester Luiz Galvão Cesar, Moisés Goldbaum, Camila Nascimento Monteiro.

---

## [Editor Report · Decision Letter 1]

14 Dec 2023

Physical activity among older adults with multimorbidity: Evidence from a population-based health survey

PONE-D-23-19647R1

Dear Dr. Ferreira,

We’re pleased to inform you that your manuscript has been judged scientifically suitable for publication and will be formally accepted for publication once it meets all outstanding technical requirements.

Kind regards,

Mathias Roberto Loch, Ph.D

Academic Editor

PLOS ONE
---

## [Editor Report · Acceptance letter]

20 Dec 2023

PONE-D-23-19647R1 

PLOS ONE

Dear Dr. Ferreira, 

I'm pleased to inform you that your manuscript has been deemed suitable for publication in PLOS ONE. Congratulations! Your manuscript is now being handed over to our production team.

Kind regards, 

on behalf of

Dr. Mathias Roberto Loch 

Academic Editor

PLOS ONE